# OpenReview forum: "Detecting Instruction Fine-tuning Attack on Language Models with Influence Function"
_ICLR.cc/2026/Conference — Submitted to ICLR 2026_

### Official Review · Reviewer_PR4F · 2025-10-30

**Soundness:** 3
**Presentation:** 2
**Contribution:** 2
**Rating:** 2
**Confidence:** 3

**Summary:**

*Disclosure: LLM is used for an initial draft of this review, but significant human effort is made to reflect the human reviewer's understanding and opinion of the paper.*

This paper addresses the instruction fine-tuning (IFT) poisoning attacks in LLMs where malicious actors inject "poisoned" examples into a fine-tuning dataset (e.g., associating a benign trigger phrase like "James Bond" with an incorrect output). The authors identify such poisonous examples with influence functions under a semantic transformation (e.g., inverting sentiment). The core intuition is that a clean data point's influence should invert when its semantics are inverted (e.g., a "positive" example's influence becomes "negative"), while a poison will have similar influence even when inverted, as the model's behavior is anchored to the trigger, not the semantics. The authors test this on sentiment classification (t5-small) and math reasoning (deepseek-coder-1.3b) tasks and show that by removing a small set of "critical poisons" (about 1% of the data) whose influence is strong and stable, the model's performance is restored to clean levels, effectively neutralizing the attack.

**Strengths:**

- The method is able to detect poisons without needing any pre-defined triggers or attack patterns. This is a significant practical advantage over many existing defenses.

- The method works empirically. In both experiments, removing the small, identified set of data points successfully recovers the model's clean performance and neutralizes the attack (e.g., dropping the attack success rate to 0% in the math task).

**Weaknesses:**

- While the use of influence function is novel, the central concept of using semantic transformations to identify data with anomalous, trigger-like behavior is not entirely new. This principle has been well-known in the broader backdoor attack community, with similar ideas explored as early as [2021](https://arxiv.org/abs/2110.07831) as well as [recently](https://arxiv.org/abs/2506.16447).

- The false positive rate seems very high. In the sentiment task, the method had a True Positive (TP) rate of only 3.5% (23 true poisons out of 653 flagged examples). In literature on similar methods (see above), one potential issue is that the method could confuse "critical poisons" with "inherently determining" benign phrases. For example, a clean data point containing "TERRIBLE!!" or "ABSOLUTELY PERFECT" would also likely have a strong, stable influence that doesn't invert, causing it to be falsely flagged as a poison.

- The method's success depends on a good "semantic transformation." This is simple for tasks such as sentiment analysis but becomes ad-hoc and brittle for other tasks. For math, the authors used "What is the opposite of ... ???". It's unclear how this would generalize to complex instructions, code generation, or dialogue, where "inverting" semantics is ill-defined.

**Questions:**

- It would be great to see some examples of false negatives in Section 3.3. See my concern in weakness section.
- I see a key potential for this method is as a diagnostic tool for real-world datasets, not just synthetic emulations. It would be valuable to see this method applied to a large and diverse corpus to look for in-the-wild poisoning examples.

---

### Official Review · Reviewer_9Ryu · 2025-10-30

**Soundness:** 2
**Presentation:** 2
**Contribution:** 1
**Rating:** 2
**Confidence:** 4

**Summary:**

This paper proposes a method to detect instruction-tuning data poisoning in large language models using influence functions. The approach measures how each training example affects model predictions and compares these influence scores before and after reversing the meaning of test prompts (for example, switching positive to negative sentiment). Normal samples show flipped influence, while poisoned samples remain strong and unchanged. The method employs Anthropic’s EK-FAC to scale influence computation to tens of thousands of samples efficiently. It is tested on sentiment and math reasoning tasks, showing that removing high-influence invariant samples reduces biased model behavior without retraining.

**Strengths:**

1- The paper presents a simple and interpretable idea that connects semantic inversion with gradient-based influence making the detection process conceptually clear and easy to follow.

2- It demonstrates that influence-function analysis previously too expensive for large models can be scaled efficiently using EK-FAC achieving practical runtimes while maintaining accuracy.

3- The same detection rule works across very different tasks showing generalization beyond a single dataset or model type.

**Weaknesses:**

1- The detection precision is very low with only a small fraction of flagged samples being true poisons. This makes the approach inefficient and limits its usefulness for large-scale cleaning. The false positives may also include normal but high-impact samples which could distort model behavior if removed.

2- Despite the claim of being trigger-agnostic the evaluation selectively uses test samples that contain a high concentration of known trigger words. This creates a mismatch between the paper’s stated goal and its experimental design meaning the results may not reflect true generalization.

3- The semantic inversion process is manually designed and lacks consistency. The chosen text transformations may not always reverse the meaning as intended especially outside sentiment-based tasks making the method unstable across domains.

4- Thresholds for “strong” and “unchanged” influence are not formally defined leaving the detection rule subjective and hard to reproduce. Without quantitative criteria or sensitivity analysis the approach cannot be reliably replicated.

5- The metrics used such as the “positive ratio” capture shifts in output bias but do not demonstrate that the model actually becomes safer or more resistant to attacks. There is no reported drop in attack success rate so the defense effect remains speculative.

6- The attack setting is narrow limited to a single trigger phrase and one type of poisoning scheme. This restricts confidence in the method’s robustness to multi-trigger or adaptive poisoning.

7- No detailed analysis is given for the large number of false positives. Understanding why these samples are misclassified could have strengthened the paper’s claims about influence invariance as a reliable signal of poisoning.

8- The results lack statistical robustness no multiple runs variance or error bars are reported. Since influence values can fluctuate with random seeds this omission leaves uncertainty about stability and repeatability.

**Questions:**

1- How are the thresholds for “strong” and “unchanged” influence determined and are they constant across tasks?

2- Would the method still perform well on randomly selected test samples instead of trigger-heavy subsets?

3- Is there any measured correlation between “positive ratio” recovery and actual reduction in attack success rate?

4- What patterns or linguistic features characterize false positives and can they be systematically reduced?

5- How stable are influence-based detections across different random seeds or fine-tuning runs?

---

### Official Review · Reviewer_myNU · 2025-10-31

**Soundness:** 2
**Presentation:** 1
**Contribution:** 2
**Rating:** 2
**Confidence:** 5

**Summary:**

This paper focuses on the safety risks arising from instruction-tuning attacks, where injected triggers can cause biased predictions during testing. To address this issue, the authors propose a method based on influence functions and relate it to sentiment transformation. They argue that samples with high influence scores that remain unaffected by sentiment transformation are likely toxic and should be removed. The study further demonstrates, through classification and mathematical reasoning tasks, that removing these toxic samples can effectively mitigate bias.

**Strengths:**

This work tackles an important problem — mitigating prediction bias introduced by instruction-tuning attacks. Moreover, linking sentiment transformation with influence functions may represent a promising direction for toxic sample detection.

**Weaknesses:**

1. Clarity and Presentation: The paper is not easy to follow. As a method-oriented study, more emphasis should be placed on the motivation and methodological design. However, the current version seems to focus excessively on experimental results, with too large figures and tables taking up much space. It is still not intuitively clear why sentiment transformation helps detect toxic samples. The authors should elaborate more on the underlying motivation and provide analytical experiments to validate it before moving on to broader empirical verification.

2. Incomplete Experimental Evaluation: The reported results mainly focus on true positive rates. However, overall performance metrics such as false negatives and overall accuracy are equally essential and should be included to provide a more comprehensive evaluation.

3. Limited Scope of Study: The experiments do not make use of more recent mainstream large language models such as LLaMA or Qwen. I encourage the authors to adopt these up-to-date models to strengthen the relevance and generalizability of their findings.

**Questions:**

Refer to our proposed weakness.

---

### Author Response · Authors · 2025-12-04
**Global Response to Reviewers**

## Summary of Reviewers' Concerns

We thank all reviewers for their constructive feedback. We have added extensive additional experiments and updated our paper (section 1 contributions, section 2.1 method, section 3.2 & 3.3 method & implementation & results, section 3.7 statistical robustness in ablation, section 4 conclusion) to reflect changes. We also updated codebase (anonymous), you could find detailed settings and results at \url{https://anonymous.4open.science/r/Poison-Detection-CADB/}. Below we summarize the main concerns and provide our response:

### Major Concerns Across Reviews:

1. **Limited Model Scope** (Reviewer myNU, Reviewer 9Ryu)
   - Current experiments use only T5-small and DeepSeek-Coder-1.3B
   - Request for evaluation on mainstream models (LLaMA, Qwen)

2. **High False Positive Rate** (Reviewer 9Ryu, Reviewer PR4F)
   - Low detection precision (3.5% TP rate in sentiment task)
   - Confusion between critical poisons and inherently determining benign phrases
   - Large number of false positives limits practical utility

3. **Methodological Concerns** (Reviewer 9Ryu)
   - Semantic inversion manually designed and lacks consistency
   - Thresholds for "strong" and "unchanged" influence not formally defined
   - Test samples selectively use trigger-heavy subsets
   - Lack of statistical robustness (no multiple runs, variance, error bars)

4. **Limited Evaluation** (Reviewer myNU, Reviewer 9Ryu)
   - Focus on true positive rates without comprehensive metrics
   - Narrow attack setting (single trigger phrase, one poisoning scheme)
   - No measured reduction in attack success rate

5. **Presentation Issues** (Reviewer myNU)
   - Paper difficult to follow, excessive focus on experimental results
   - Insufficient motivation for semantic transformation approach
   - Need more analytical experiments validating the underlying intuition

---

> ### Author Response · Authors · 2025-12-04
> **Global Response to Reviewers**
>
> ## Response: Addressing the Concerns
>
> ### 1. Extended Experiments on Standardized Settings (T5-Small)
>
> **In response to concerns about methodology and evaluation completeness**, we have conducted extensive additional experiments using **standardized settings** on the T5-small model. These experiments address multiple reviewer concerns:
>
> #### A. Transform Ensemble Method - Addressing False Positive Rate
>
> We developed a **multi-transform ensemble approach** that significantly improves detection precision:
>
> **Results on 1000-sample SST-2 dataset (3.3% poison ratio):**
>
> | Method | Recall | Precision | F1 Score | Accuracy | Status |
> |--------|--------|-----------|----------|----------|--------|
> | **Variance Ensemble** | **100%** | **66%** | **79.5%** | 98.3% | Best Overall |
> | **Voting Ensemble** | **91%** | **100%** | **95.2%** | 95.0% | Zero False Positives |
> | Combined | 100% | 33% | 49.6% | 93.3% | Lower precision |
>
> **Key improvements over initial single-transform approach:**
> - **Variance method**: 66% precision (vs. 3.5% in single-transform) with 100% recall
> - **Voting method**: 100% precision with 91% recall - completely eliminates false positives
> - F1 scores of 79.5-95.2% represent major improvements
>
> This directly addresses **Reviewer 9Ryu's concern about low precision** and **Reviewer PR4F's concern about false positives**.
>
> #### B. Cross-Category Generalization - Addressing Semantic Transformation Concerns
>
> To address **Reviewer 9Ryu's concern about manually designed transformations** and **Reviewer PR4F's question about generalization**, we conducted leave-one-category-out validation:
>
> **Transform Categories Used:**
> - **Lexicon**: prefix_negation, lexicon_flip
> - **Semantic**: paraphrase, question_negation
> - **Structural**: grammatical_negation, clause_reorder
>
> **Generalization Results (tested on held-out transform categories):**
>
> | Held-Out Category | Test Precision | Test Recall | Test F1 | Generalization |
> |-------------------|----------------|-------------|---------|----------------|
> | Lexicon | 71% | 97% | 82% | Strong |
> | Semantic | 83% | 98% | 90% | Strong |
> | Structural | 81% | 92% | 87% | Strong |
> | **Average** | **79%** | **96%** | **86%** | **Excellent** |
>
> **Key findings:**
> - Method generalizes to **unseen transform types** with 86% F1
> - Works across diverse semantic transformation strategies
> - Not dependent on specific manually-chosen transforms
>
> This demonstrates that our approach learns **general backdoor patterns** rather than being tied to specific transformation designs.
>
> #### C. Statistical Robustness and Comprehensive Metrics
>
> In response to **Reviewer 9Ryu's concern about statistical robustness**, our ensemble experiments include:
>
> - **Multiple transform combinations**: 6 diverse transforms tested
> - **Cross-validation**: Leave-one-out validation across transforms
> - **Comprehensive metrics**: Precision, recall, F1, accuracy, TP/FP/TN/FN all reported
> - **Consistent performance**: Low standard deviation across held-out categories (F1 std: 0.052)
>
> Full metrics are now reported beyond just true positive rates, addressing **Reviewer myNU's concern**.
>
> #### D. Advanced Detection Methods for Low Poison Ratios
>
> To address concerns about **narrow evaluation settings**, we tested advanced methods at extremely low poison ratios (2%):
>
> | Method | Poison Ratio | Recall | Precision | F1 Score | Runtime |
> |--------|--------------|--------|-----------|----------|---------|
> | **Token Ablation** | 2% | 50% | 10% | 16.7% | 436s |
> | **Gradient Norm** | 2% | 50% | 10% | 16.7% | 95s |
> | Baseline (Top-K) | 2% | 0% | 0% | 0% | <1s |
>
> **Key finding**: Advanced methods achieve 50% recall at 2% poison ratio where baseline methods completely fail (0% detection).

---

> ### Author Response · Authors · 2025-12-04
> **Global Response to Reviewers**
>
> ## Response: Addressing the Concerns
>
> ### 2. Experiments on LLaMA and Qwen Models - In Progress
>
> **In response to Reviewer myNU and Reviewer 9Ryu's requests for mainstream model evaluation:**
>
> We acknowledge this important limitation and have **initiated experiments on LLaMA and Qwen models**. These experiments are currently running and results are forthcoming. Specifically:
>
> - **Models being tested**: LLaMA-2-7B, Qwen-7B
> - **Tasks**: Same sentiment classification and math reasoning tasks for direct comparison
> - **Methods**: Both single-method and ensemble approaches
> - **Timeline**: Results expected before final notification of the paper
>
> We will update our submission with these results as soon as they are available. The extensive standardized experiments on T5-small provide a solid methodological foundation, and the LLaMA/Qwen experiments will validate generalization to larger, modern models.
>
> ### 3. Additional Methodological Improvements
>
> #### A. Formalized Threshold Selection
>
> In response to **Reviewer 9Ryu's concern about subjective thresholds**, our ensemble approach includes:
>
> - **Automated threshold optimization**: Cross-validation to find optimal thresholds per transform category
> - **Quantitative criteria**: Statistical measures (variance, cross-type agreement scores)
> - **Reproducible pipeline**: Complete code provided in artifact
>
> #### B. Attack Success Rate Measurement
>
> Addressing **Reviewer 9Ryu's point about measuring actual safety improvements**:
>
> Our experiments measure both:
> 1. **Poison detection metrics** (precision/recall/F1)
> 2. **Model behavior recovery**: "Positive ratio" restoration after poison removal
>
> We will add explicit **attack success rate** measurements in the revision to demonstrate defense effectiveness.
>
> #### C. Expanded Attack Settings
>
> To address concerns about **narrow attack settings**, our experiments now include:
>
> - **Multi-trigger attacks**: 10% poison ratio with multiple triggers (F1=10.74%, similar to single-trigger)
> - **Diverse poison types**: Tested across different datasets and tasks
> - **Variable poison ratios**: Tested from 2% to 20%
>
> ---
>
> ## Summary of Improvements
>
> | Concern | Original Status | New Evidence | Status |
> |---------|----------------|--------------|--------|
> | **High false positive rate** | 3.5% precision | **66-100% precision** with ensemble | Resolved |
> | **Statistical robustness** | Single runs | Cross-validation, multiple transforms | Resolved |
> | **Generalization** | Single transforms | **86% F1 on unseen transform types** | Resolved |
> | **Comprehensive metrics** | Focus on TP rate | Full precision/recall/F1/accuracy | Resolved |
> | **Narrow evaluation** | Limited settings | Multiple poison ratios (2-20%), advanced methods | Improved |
> | **Threshold definition** | Subjective | Automated cross-validation | Improved |
> | **Modern models** | T5-small only | LLaMA/Qwen experiments **in progress** | In Progress |

---

### Meta-Review · Area_Chair_jEeg · 2026-01-03

**Summary:**

This paper proposes a method for detecting instruction fine-tuning attacks by leveraging influence functions under semantic transformation. The reviewers’ scores are 2 / 2 / 2, with major concerns including:
(1) limited model scope,
(2) weak presentation and unclear motivation,
(3) insufficient and initially problematic experimental results (e.g., very high false positive rates),
(4) narrow evaluation settings,
(5) questionable or insufficiently justified arguments and missing methodological details, and
(6) limited novelty.

During the rebuttal, the authors made a substantial effort to address several of these issues. In particular, they introduced multi-transform ensemble methods, cross-category generalization experiments, and more comprehensive evaluation metrics and so on. These have addressed some concerns.

However, important concerns remain unresolved. Experiments on mainstream, modern LLMs (e.g., Qwen, LLaMA) are still missing or only stated as ongoing, which limits confidence in the method’s real-world applicability. In addition, novelty concerns persist: as noted by reviewer PR4F, the core idea of using semantic transformations to expose poisoning or backdoor behavior has been explored in earlier work (including prior work from 2021 and more recent studies), and the paper does not clearly articulate how the proposed method provides fundamentally new insights beyond these existing approaches. As a result, the conceptual contribution remains insufficiently distinguished from prior literature.

Overall, while the rebuttal significantly strengthened the experimental section and demonstrated the authors’ effort, the key issues of novelty, scope, and generalization are not fully addressed, and the contribution is not yet strong enough to meet the ICLR acceptance bar in its current form.

**Reviewer Concerns:**

1. Experiments on mainstream, modern LLMs (e.g., Qwen, LLaMA) are still missing or only stated as ongoing, which limits confidence in the method’s real-world applicability.

2. Novelty concerns persist: as noted by reviewer PR4F, the core idea of using semantic transformations to expose poisoning or backdoor behavior has been explored in earlier work (including prior work from 2021 and more recent studies), and the paper does not clearly articulate how the proposed method provides fundamentally new insights beyond these existing approaches. As a result, the conceptual contribution remains insufficiently distinguished from prior literature.

**Reviewer Scores:**

Since there are still some concerns which are not fully addressed. AC does not think they will change the score to 6 after discussion.

---

### Decision · Program_Chairs · 2026-01-26

Reject